# Resilience Protects Nurses from Workplace Gaslighting and Quiet Quitting, and Improves Their Work Engagement: A Cross-Sectional Study in Greece

**DOI:** 10.3390/healthcare13162064

**Published:** 2025-08-20

**Authors:** Ioannis Moisoglou, Aglaia Katsiroumpa, Olympia Konstantakopoulou, Ioanna V. Papathanasiou, Ioanna Prasini, Maria Rekleiti, Petros Galanis

**Affiliations:** 1Department of Nursing, University of Thessaly, 41500 Larissa, Greece; iomoysoglou@uth.gr (I.M.); iopapathanasiou@uth.gr (I.V.P.); 2Clinical Epidemiology Laboratory, Faculty of Nursing, National and Kapodistrian University of Athens, 11527 Athens, Greece; aglaiakat@nurs.uoa.gr; 3Center for Health Services Management and Evaluation, Faculty of Nursing, National and Kapodistrian University of Athens, 11527 Athens, Greece; olykonstant@nurs.uoa.gr; 4Palliative Care Galilee, 19004 Spata, Greece; iprasini@galilee.gr; 5Andreas Syggros Hospital of Cutaneous and Venereal Diseases, 16121 Athens, Greece; proistep@syggros.gr

**Keywords:** resilience, nurses, workplace gaslighting, Gaslighting at Work Scale, quiet quitting, work engagement

## Abstract

**Background:** Although gaslighting is an alarming issue, the literature on predictors of this phenomenon is scarce. **Objective:** To examine the association between resilience and gaslighting in the workplace, quiet quitting, and work engagement among nurses. **Methods:** We conducted a cross-sectional study in Greece during December 2024. We used the Brief Resilience Scale (BRS) to measure levels of resilience in our sample. Moreover, we used the Gaslighting at Work Scale (GWS), the Quiet Quitting Scale, and the Utrecht Work Engagement Scale-3 to measure workplace gaslighting, quiet quitting, and work engagement, respectively. **Results:** The study population included 462 nurses with a mean age of 36.80 years. We found that resilience protected nurses from workplace gaslighting. After adjusting gender, age, educational level, and work experience, a negative association was found between resilience and GWS score (*p* < 0.001), loss of self-trust (*p* < 0.001), and abuse of power (*p* < 0.001). Moreover, our multivariable analysis identified a negative association between resilience and Quiet Quitting Scale score (*p* < 0.001), detachment (*p* < 0.001), lack of initiative (*p* < 0.001), and lack of motivation (*p* < 0.001). Additionally, we identified a positive relationship between resilience and work engagement (*p* < 0.001). **Conclusions:** Our findings suggested the protective role of resilience against gaslighting in the workplace and quiet quitting in nurses. Moreover, we found that resilience improved nurses’ work engagement. However, the cross-sectional nature of this study cannot imply causality between the study variables, and, thus, further studies are required to clarify the association between resilience, workplace gaslighting, quiet quitting, and work engagement.

## 1. Introduction

The nursing work environment is exceptionally challenging and stressful. Nurses frequently deal with trauma and terminally ill patients, experience workplace violence, and lack organizational support and essential resources [1,2]. Within this context, resilience is an important asset for nurses to cope and continue their work effectively. Resilience can be defined as “*a complex and dynamic process which, when present and sustained, enables nurses to adapt positively to workplace stressors, avoid psychological harm, and continue to provide safe, high-quality patient care*” [3]. Nurses exhibiting elevated resilience levels encounter reduced prevalence of burnout, secondary traumatic stress, depression, and turnover, concurrently displaying heightened self-esteem, life, and job satisfaction [4,5,6]. Furthermore, resilience serves as a protective factor for both patients and nursing staff by enhancing the safety climate and the overall performance of nurses, thereby reducing the incidence of occupational accidents [7,8]. The benefits of resilience also enhance the overall quality of patient care, as resilience mitigates the adverse effects of compassion fatigue among nurses on care quality [9]. Throughout the COVID-19 pandemic, when nurses faced a significant number of challenges, resilience emerged as a protective factor mitigating the burnout resulting from the pandemic’s effects on their personal and professional lives [10]. Alongside biological components, numerous environmental factors, especially resources, also influence resilience [11]. Nursing leadership and management have a crucial role in obtaining essential resources for cultivating resilience. By engaging in behaviors that enhance social relationships, promote positivity, leverage nurses’ strengths, cultivate their growth, encourage self-care, support mindfulness practices, and express compassion, nursing leaders can strengthen the resilience of their staff [12]. Nurses who received organizational support and the opportunity to engage in policy and procedure development during the COVID-19 pandemic exhibited elevated levels of resilience [13].

However, nurses often not only fail to receive organizational support but also fall victim to psychological manipulation by their supervisors through gaslighting. The term “gaslighting” originated in a 1944 film centered on a couple’s relationship. Subsequent reports have highlighted cases in which individuals without underlying mental health disorders were subjected to gaslighting, resulting in their involuntary institutionalization by close associates or relatives, motivated by financial gain or other personal benefits. Later, incidents emerged of gaslighting victims being hospitalized, and ultimately, there are accounts of elderly individuals being subjected to gaslighting by care home personnel or relatives in their residences for financial exploitation [14]. Stern, author and psychoanalyst, provides the following definition of gaslighting in the introduction to the second edition of her book, “*Gaslighting, is a type of emotional manipulation in which a gaslighter tries to convince you that you’re misremembering, misunderstanding, or misinterpreting your own behaviors or motivations, thus creating doubt in your mind that leaves you vulnerable or confused*” [15]. The aims for gaslighting conduct may encompass material and financial advantages, albeit they are not confined to these aspects. Additional motivations for this activity may encompass the gaslighter’s insecurity, the desire to be right, the desire to exert control, the desire to eliminate any potential for disagreement, and the urge to alleviate their anxiety through projective identification [14,16,17]. Gaslighters employ multiple tactics to attain their objectives, including denial (refusing to recognize truths despite evidence, substituting dismissive statements for dialogue), deception, dismissal, minimization, behavioral inconsistency, isolation, and coercion [17]. The consequences of gaslighting are extensive and primarily impact the mental well-being of victims, encompassing withdrawal from interpersonal and social contexts, anxiety, guilt, depression, grief, suicidal ideation and actions, and compromised mental health. Experiencing gaslighting behavior heightens the likelihood of developing post-traumatic stress disorder (PTSD), which may manifest as flashbacks, nightmares, anxiety, social withdrawal, apathy, hypervigilance, insomnia, or recurrent anger outbursts [18]. Studies on gaslighting in the workplace, although limited, have highlighted its effects on employees, including its negative impact on job embeddedness, motivation, and affective organizational commitment [19,20]. In the healthcare sector, nurses who are victims of gaslighting are more likely to experience burnout, express an intention to leave their job, and have their career entrenchment and agility negatively affected [21,22,23].

In recent years, the concept of work engagement has attracted increasing scholarly interest. Evidence from a large-scale international survey, encompassing more than 200,000 employees across 160 countries and spanning both private and public sectors, indicated that only one in five employees demonstrates genuine engagement in their work [24]. Engaged employees exhibit considerable energy and mental resilience, display enthusiasm and pride in their work, draw inspiration from it, and become so engrossed that they lose track of time while working [25]. Engaged nurses exhibit heightened job satisfaction and enhanced performance, manage daily work demands more efficiently, and express reduced inclinations to leave their positions [26,27,28]. Simultaneously, nurses work engagement serves as a predictor of care quality [28,29]. As in the case of resilience, supportive nursing leadership that responds to nurses’ needs can enhance work engagement, while a poor work environment limits it [30,31]. Also, when nurses cultivate a high degree of resilience, they seem to exhibit greater engagement in their work [32,33].

Quiet quitting is a work behavior that has been progressively prevalent since the COVID-19 pandemic. Initially revealed via a brief video on social media, it soon became evident that a significant proportion of employees were already engaging in this behavior [34,35]. Employees who choose quiet quitting do not formally resign but restrict their performance to the minimum required to evade termination. They fail to articulate novel concepts, refrain from working beyond regular hours, and typically have a passive disposition toward their tasks [35,36]. Despite the apparent paradox of nurses opting to quiet quitting, considering the demanding and continuous nature of patient care, findings indicate that a significant proportion of nurses engage in this work behavior more frequently than other healthcare professionals [37,38,39]. Job burnout and dissatisfaction, excessive workload, insufficient recognition of nurses’ efforts, bullying, workplace conflicts, perceived injustice, and stress have been identified as factors associated with the emergence of this work behavior [38,40,41,42,43]. Conversely, perceived workplace support and innovation support mitigate this phenomenon [40,44]. Nurses who choose quiet quitting are not “comfortable” with this condition for the rest of their working lives. Essentially, quiet quitting is an attempt to protect themselves in a particularly demanding and unsupportive work environment, while at the same time seeking a way out, as they have higher turnover intention rates [45].

Nursing personnel in Greece face challenges similar to those encountered by nurses worldwide. Understaffing and excessive workload constitute the two most significant challenges for nurses employed in hospital settings [30,43]. According to the Organisation for Economic Co-operation and Development (OECD), Greece ranks among the lowest countries in terms of the number of nurses per 1000 inhabitants, with figures remaining unchanged over the past decade [46]. Major issues affecting the occupational well-being of nurses in Greece include high levels of burnout, job dissatisfaction, intention to leave the profession, low work engagement, and a poor work environment [30,45,47]. Within this professional context, nurses’ resilience is deemed fundamental to ensuring the continued delivery of high-quality and safe healthcare services.

To the best of our knowledge, this is the first study that investigated the association between resilience and gaslighting in the workplace and quiet quitting, and one of the few that explored resilience and work engagement among nurses.

## 2. Materials and Methods

### 2.1. Study Design

A cross-sectional study was implemented in Greece, with data gathered through an online survey in December 2024. The study questionnaire was digitized using Google Forms and distributed via nurses’ social media groups on Facebook and Instagram, as well as through LinkedIn messages. Nurses’ groups were institutional networks and networks of professional associations of nurses. This method resulted in a convenience sample. Our inclusion criteria were the following: (a) participants were required to be clinical nurses in healthcare facilities, (b) nurses should be subordinates and not supervisors, (c) nurses should have at least one year of work experience, and (d) nurses should consent to participate in our study. Before nurses completed the study questionnaire, we asked them if they had been working as clinical nurses for at least one year. Nurses with positive answers were then allowed to fill in the study questionnaire. The study adhered to the Strengthening the Reporting of Observational studies in Epidemiology (STROBE) guideline [48].

We used G*Power v.3.1.9.2 to calculate our sample size. We included one predictor (resilience) and four confounders (gender, age, educational level, and work experience) in our models. Thus, considering an anticipated effect size of 0.03 between resilience and outcomes (workplace gaslighting, quiet quitting, and work engagement), a statistical power of 95%, and a margin of error of 5%, the sample size was estimated to be 436 nurses.

### 2.2. Measurements

We measured the following demographic data: gender (males or females), age (continuous variable), educational level (MSc/PhD diploma), and work experience (continuous variable).

Resilience was evaluated using the Brief Resilience Scale (BRS), a six-item instrument with items like “I tend to bounce back quickly after hard times” and “I have a hard time making it through stressful events” [49]. Responses were recorded on a 5-point Likert scale ranging from strongly disagree (1) to strongly agree (5). The BRS total score spans from 1 to 5, with higher scores indicating greater resilience. The validated Greek version of the BRS was utilized [50], yielding a Cronbach’s alpha of 0.805 in our study.

The Gaslighting at Work Scale (GWS) was employed to assess workplace gaslighting among nurses [51]. This 11-item scale includes statements such as “In the last six months, your supervisor denies saying things that you remember him/her saying”, “In the last six months, your supervisor lies to you”, and “In the last six months, your supervisor makes you depend on him/her for making decisions about your work”. The GWS comprises two factors: “loss of self-trust” (five items) and “abuse of power” (six items). Responses are recorded on a 5-point Likert scale from never (1) to always (5). The GWS score and its two factors are calculated as the average of all answers, ranging from 1 to 5, with higher scores indicating more frequent gaslighting behaviors from supervisors. The Greek version of the GWS was used, demonstrating a Cronbach’s alpha of 0.940. The “loss of self-trust” and “abuse of power” factors showed Cronbach’s alpha values of 0.902 and 0.904, respectively. The Greek version of the GWS is a reliable and valid tool since a study with 400 participants confirmed the two-factor 11-item structure of the scale through factor analysis [51]. Additionally, the GWS showed very good concurrent validity through moderate to high correlations with four other valid scales. Cronbach’s alpha and McDonald’s Omega for the GWS were 0.939 and 0.949, respectively. Additionally, the test-retest study showed very high intraclass correlation coefficients and Cohen’s kappa values [51].

To assess quiet quitting among nurses, we employed the Quiet Quitting Scale [52]. This instrument comprises nine items, such as “I do the basic or minimum amount of work without going above and beyond”, “I take as many breaks as I can”, and “I do the basic or minimum amount of work without going above and beyond”. Responses are recorded on a 5-point Likert scale, ranging from 1 (strongly disagree/never) to 5 (strongly agree/always). The Quiet Quitting Scale encompasses three factors: “detachment” (four items), “lack of initiative” (three items), and “lack of motivation” (two items). Each factor’s score is calculated as the mean of its item responses, resulting in a range from 1 to 5. Higher scores indicate increased levels of quiet quitting. We utilized the validated Greek version of the Quiet Quitting Scale. In our study, the Cronbach’s alpha for the Quiet Quitting Scale was 0.851. Additionally, Cronbach’s alpha for the factors “detachment,” “lack of initiative,” and “lack of motivation” was 0.812, 0.763, and 0.797, respectively. The Greek version of the Quiet Quitting Scale has already proven reliable and valid in a sample of nurses [39]. In short, confirmatory factor analysis confirmed the three-factor, nine-item structure of the Quiet Quitting Scale since all goodness-of-fit statistics have acceptable values. Also, all Cronbach’s alpha and McDonald’s Omega values were higher than 0.70. Moreover, Cohen’s kappa for the nine items ranged from 0.840 to 0.947, while intraclass correlation coefficients for the Quiet Quitting Scale and the three subfactors ranged from 0.972 to 0.988.

Work engagement was measured using the Utrecht Work Engagement Scale-3 (UWES-3) [39,53]. This three-item tool includes questions like “At my work, I feel bursting with energy”, with responses on a seven-point Likert scale from never (0) to every day (6). The UWES-3 mean score ranges from 0 to 6, with higher scores signifying greater work engagement. The validated Greek version of the UWES-3 was utilized, yielding a Cronbach’s alpha of 0.765 in this study [54].

### 2.3. Ethical Issues

Our study adhered to the Declaration of Helsinki guidelines [55]. The Ethics Committee of the Faculty of Nursing, National and Kapodistrian University of Athens approved our study protocol (approval number: 15, 9 December 2024). Data collection was conducted anonymously and voluntarily. Participants were informed about the study’s aim and design and provided their informed consent.

### 2.4. Statistical Analysis

We present categorical variables as numbers and percentages, while continuous variables are described using mean, standard deviation (SD), median, and interquartile range. The Kolmogorov–Smirnov test and Q-Q plots were used to examine the distribution of continuous variables, which were found to follow a normal distribution. Resilience was the independent variable, while workplace gaslighting, quiet quitting, and work engagement were the dependent variables. Demographic variables (gender, age, educational level, and work experience) were considered potential confounding factors. To identify associations between resilience, workplace gaslighting, quiet quitting, and work engagement, we conducted simple and multivariable linear regression analyses. We first performed simple linear regression analysis, followed by the construction of a final multivariable model. This model eliminated confounders to estimate the independent effect of resilience on the dependent variables. Age and work experience showed high correlation (Pearson’s correlation coefficient = 0.946, *p*-value < 0.001). To avoid multicollinearity issues in the multivariable models, we included work experience rather than age in these models. We presented unadjusted and adjusted coefficients beta, 95% confidence intervals (CI), standardized coefficients beta, and *p*-values. Moreover, we used variance inflation factors (VIFs) to assess multicollinearity in the multivariable models. A VIF greater than 5 indicates multicollinearity between independent variables. Additionally, we examined histograms of the residuals to check for multivariable normality. We examined scatterplots of residuals versus predicted values to check for homoscedasticity and linearity [56]. There was no missing data. *p*-values less than 0.05 were considered statistically significant. We used the IBM SPSS 28.0 (IBM Corp. Released 2021. IBM SPSS Statistics for Windows, Version 28.0. Armonk, NY, USA: IBM Corp.) for the analysis.

## 3. Results

### 3.1. Demographic Characteristics

Table 1 presents the demographic profile of the nurses in the study. The sample consisted of 462 nurses, predominantly female (85.3%). The average age was 36.8 years (SD: 10.49), with a median of 35.5 years (interquartile range: 16). More than half of nurses possessed an MSc/PhD diploma (54.3%). The mean work experience was 12.84 years (SD: 10.11), with a median of 10 years (interquartile range: 16).

### 3.2. Study Scales

Table 2 outlines the descriptive statistics for the study scales. The mean resilience score (3.31, SD: 0.68) indicated a moderate level of resilience among our nurses.

The GWS mean score was 2.57 (SD: 0.95), with “loss of self-trust” and “abuse of power” factors scoring 2.25 and 2.84, respectively. This indicates moderate levels of gaslighting behaviors from supervisors, with abuse of power being more prevalent than loss of self-trust. Quiet quitting levels were moderate (Quiet Quitting Scale mean: 2.42), with lack of motivation (mean: 2.81) occurring more frequently than lack of initiative (mean: 2.44) and detachment (mean: 2.22). Work engagement was moderate, with a UWES-3 mean score of 3.48 (SD: 1.46).

### 3.3. Association Between Resilience and Workplace Gaslighting

Table 3 reveals that resilience protected nurses from workplace gaslighting. After adjusting for gender, age, educational level, and work experience, a negative association was found between resilience and GWS score (b = −0.427, 95% CI = −0.551 to −0.304, *p* < 0.001), loss of self-trust (b = −0.387, 95% CI = −0.514 to −0.260, *p* < 0.001), and abuse of power (b = −0.461, 95% CI = −0.595 to −0.327, *p* < 0.001). Thus, more resilient nurses experienced lower levels of gaslighting from their supervisors.

Appendix A indicates multivariable normality for the multivariable model with loss of self-trust as the dependent variable since the residuals followed a normal distribution. Appendix A indicates homoscedasticity and linearity of the multivariable model, with loss of self-trust as the dependent variable. VIFs for the final multivariable model ranged from 1.040 to 1.060, indicating an absence of multicollinearity between independent variables.

Appendix A refer to a multivariable model with abuse of power as the dependent variable. Figures indicate normality, homoscedasticity, and linearity of the multivariable model. Moreover, VIFs for the final multivariable model ranged from 1.040 to 1.060, indicating an absence of multicollinearity between independent variables.

Appendix A denotes multivariable normality for the multivariable model with workplace gaslighting score as the dependent variable since the residuals followed a normal distribution. Also, we found that the multivariable model shows homoscedasticity and linearity (Appendix A). There was no multicollinearity between independent variables since VIFs ranged from 1.040 to 1.060.

### 3.4. Association Between Resilience and Quiet Quitting

Linear regression analysis results for resilience and quiet quitting are presented in Table 4. After controlling the confounding factors, a negative relationship was observed between resilience and quiet quitting. In particular, we found a negative association between resilience and Quiet Quitting Scale score (b = −0.324, 95% CI = −0.415 to −0.232, *p* < 0.001), detachment (b = −0.216, 95% CI = −0.325 to −0.107, *p* < 0.001), lack of initiative (b = −0.419, 95% CI = −0.535 to −0.303, *p* < 0.001), and lack of motivation (b = −0.396, 95% CI = −0.521 to −0.271, *p* < 0.001). Thus, our findings suggested the protective role of resilience against quiet quitting among our nurses.

Appendix A refer to a multivariable model with detachment as the dependent variable. There was evidence for normality, homoscedasticity, and linearity of the multivariable model.

Appendix A indicates multivariable normality for the multivariable model with lack of initiative as the dependent variable since the residuals followed a normal distribution. Also, we found that the multivariable model shows homoscedasticity and linearity (Appendix A).

There was multivariable normality for the multivariable model with lack of motivation as the dependent variable (Appendix A). Also, a scatterplot of residuals versus predicted values with lack of motivation as the dependent variable indicated homoscedasticity and linearity of the multivariable model (Appendix A).

Appendix A indicates multivariable normality for the multivariable model with Quiet Quitting Scale score as the dependent variable since the residuals followed a normal distribution. Appendix A indicates homoscedasticity and linearity of the multivariable model, with Quiet Quitting Scale score as the dependent variable.

VIFs for all final multivariable models ranged from 1.040 to 1.060, indicating an absence of multicollinearity between independent variables in all cases.

### 3.5. Association Between Resilience and Work Engagement

Our multivariable model identified a positive relationship between resilience and work engagement score (b = 0.506, 95% CI = 0.312 to 0.700, *p* < 0.001). Therefore, our results suggested that resilience improves nurses’ work engagement. Table 5 shows the linear regression analysis with work engagement as the dependent variable.

Appendix A indicates multivariable normality for the multivariable model with work engagement score as the dependent variable since the residuals followed a normal distribution. Also, we found that the multivariable model shows homoscedasticity and linearity (Appendix A). There was no multicollinearity between independent variables since VIFs ranged from 1.040 to 1.060.

## 4. Discussion

This study is the first to demonstrate the substantial negative correlation between nurses’ resilience and the phenomena of workplace gaslighting and quiet quitting, while also underscoring the positive influence of resilience on work engagement. Furthermore, the results revealed moderate levels of both quiet quitting and work engagement among the participants.

Various forms of detrimental leadership, such as abusive, toxic, and manipulative leadership styles, including gaslighting, exercised by nurse supervisors affect a significant proportion of nursing staff and have a considerable impact on their mental health and work behavior [57,58]. Since such behaviors often go unnoticed by upper management, either due to underreporting or the administration’s inability to effectively address them, nurses must be equipped with the capacity to cope with these challenges. One such capacity is resilience, which enables nurses to respond to, adapt to, and recover effectively from difficult, stressful, or traumatic situations, including those arising from harmful leadership practices. When nurses are exposed to incidents of violence or inappropriate behavior, regardless of the source, their level of resilience may moderate the negative effects of such experiences [59,60]. Resilience is an important skill for nurse managers as well; those with high levels of resilience are more likely to exhibit empowering behaviors toward their staff [61]. When nurse managers adopt leadership styles such as authentic, exemplary, ethical, and transformational, they are better positioned to support nurses’ resilience in highly demanding work environments, such as those experienced during the COVID-19 pandemic [62].

Quiet quitting and diminished work engagement have become increasingly recognized as prominent workplace behaviors across diverse professional sectors, including healthcare. While recent discourse within the health sector has largely emphasized patient outcomes, particularly safety and quality of care, it is essential to highlight that frontline healthcare providers, and especially nursing staff, constitute the cornerstone of health systems. Their active involvement is fundamental to achieving optimal outcomes. Elevated levels of work engagement combined with minimal tendencies toward quiet quitting among nurses are linked to lower turnover intentions, enhanced professional performance, and superior quality of care delivery [27,45,63]. Increased levels of resilience among nurses are associated with a corresponding enhancement in their work engagement [64,65]. Given that patient care is characterized by high levels of stress, nurses with elevated levels of work engagement and resilience experience a reduced impact of stress on the development of occupational burnout [66], while also exhibiting better mental health outcomes [67].

Nursing administration plays a critical role in fostering nurses’ resilience. Administrative strategies that include the provision of formal education programs, social support, and meaningful recognition have been associated with enhanced resilience [68]. Additional approaches, such as strengthening professional ability, promoting shared governance, encouraging teamwork and mutual support among colleagues, and supporting nursing staff development (professional, practice-based, and personal), have also been identified as significant predictors of resilience development among nurses [68,69,70]. The concepts of gaslighting and quiet quitting are relatively recent in the workplace. It is highly likely that both nurses and, more importantly, their management teams are not yet familiar with the specific characteristics of these phenomena. Therefore, awareness-raising and targeted training for individuals in positions of responsibility are essential to enable the timely recognition of such workplace behaviors and the implementation of practices aimed at their reduction and eventual elimination. Furthermore, conducting studies with large samples of nurses across different countries and workplace settings will provide valuable data regarding the prevalence of these phenomena and the factors that reinforce them, thereby informing the development of appropriate managerial interventions.

Our study faced various limitations. Primarily, the cross-sectional nature of our study prevented us from establishing causality between resilience, workplace gaslighting, quiet quitting, and work engagement. Additionally, while we met the minimum sample size requirements, our use of convenience sampling introduced selection bias. For example, our participants were predominantly female, necessitating future studies with random, more representative nursing samples. Furthermore, as our study was conducted in a European country, additional research in diverse cultural contexts is needed to further explore the association between resilience, workplace gaslighting, quiet quitting, and work engagement. Although we employed multivariable models to control for several confounding factors, other potential confounders such as public or private sector employment, clinical settings, and personality traits of supervisors or subordinates were not accounted for. Future research should address these additional confounders to strengthen our findings. Additionally, self-selection bias is probable in our study since we employed an online survey to recruit our nurses. It is probable that levels of social media use were lower among older nurses with more years of experience. For instance, the mean age of our nurses was low (36.8 years). Future studies should include more representative samples to further examine the association between resilience, workplace gaslighting, quiet quitting, and work engagement. Lastly, despite using validated instruments (BRS, GWS, Quiet Quitting Scale, and UWES-3) to evaluate our study variables, the self-reported nature of these tools may have introduced information bias into our study.

## 5. Conclusions

Nurses are consistently confronted with challenges and adversities in their professional environment. To manage these effectively and sustain their clinical performance, the presence of resilience is crucial. Findings from the present study indicate that resilience functions as a protective mechanism, mitigating the risk of quiet quitting and reduced work engagement, while simultaneously safeguarding nursing staff against the detrimental impact of workplace gaslighting.

However, as we mentioned above, this study faces several limitations, and, thus, we cannot infer a causal relationship between resilience, workplace gaslighting, quiet quitting, and work engagement. Further studies should reduce bias, and, therefore, improve our ability to extract valid results. As an increasing number of nurses report disengagement and adopt quiet quitting behaviors, factors that negatively impact the quality of care, there is a pressing need for healthcare organizations to strengthen nurses’ resilience. Enhancing resilience should be regarded as a critical organizational intervention aimed at improving work engagement and mitigating the prevalence of quiet quitting among nursing professionals.

## Figures and Tables

**Table 1 healthcare-13-02064-t001:** Demographic characteristics of nurses (N = 462).

Characteristics	N	%
Gender		
Males	68	14.7
Females	394	85.3
Age (years) ^a^	36.80	10.49
MSc/PhD diploma		
No	211	45.7
Yes	251	54.3
Work experience (years) ^a^	12.84	10.11

^a^ mean, standard deviation.

**Table 2 healthcare-13-02064-t002:** Descriptive statistics for the study scales (N = 462).

Scale	Mean	Standard Deviation	Median	Interquartile Range
Resilience	3.31	0.68	3.33	1.00
Gaslighting at Work Scale	2.57	0.95	2.45	1.55
Loss of self-trust	2.25	0.97	2.00	1.60
Abuse of power	2.84	1.03	2.83	1.67
Quiet Quitting Scale	2.42	0.71	2.33	0.89
Detachment	2.22	0.82	2.00	1.00
Lack of initiative	2.44	0.90	2.33	1.33
Lack of motivation	2.81	0.96	2.50	1.50
Utrecht Work Engagement Scale-3	3.48	1.46	3.67	2.67

**Table 3 healthcare-13-02064-t003:** Linear regression models with workplace gaslighting as the dependent variable (N = 462).

Dependent Variable Independent Variable	Univariate Models				Multivariable Model ^a^			
	Unadjusted Coefficient Beta	95% CI for Beta	*p*-Value	Adjusted Coefficient Beta	95% CI for Beta	Standardized Coefficient Beta	*p*-Value	VIF
Loss of self-trust ^b^								
Resilience	−0.376	−0.502 to −0.249	<0.001	−0.387	−0.514 to −0.260	−0.271	<0.001	1.040
Abuse of power ^c^								
Resilience	−0.461	−0.593 to −0.328	<0.001	−0.461	−0.595 to −0.327	−0.304	<0.001	1.040
Workplace gaslighting ^d^								
Resilience	−0.422	−0.544 to −0.300	<0.001	−0.427	−0.551 to −0.304	−0.305	<0.001	1.040

^a^ Multivariable models are adjusted for gender, age, educational level, and work experience. ^b^ R^2^ for the multivariable model = 8.5%, *p*-value for ANOVA < 0.001. ^c^ R^2^ for the multivariable model = 10.7%, *p*-value for ANOVA < 0.001. ^d^ R^2^ for the multivariable model = 10.8%, *p*-value for ANOVA < 0.001. CI: confidence interval, VIF: variance inflation factor.

**Table 4 healthcare-13-02064-t004:** Linear regression models with quiet quitting as the dependent variable (N = 462).

Dependent Variable Independent Variable	Univariate Models				Multivariable Model ^a^			
	Unadjusted Coefficient Beta	95% CI for Beta	*p*-Value	Adjusted Coefficient Beta	95% CI for Beta	Standardized Coefficient Beta	*p*-Value	VIF
Detachment ^b^								
Resilience	−0.181	−0.290 to −0.072	<0.001	−0.216	−0.325 to −0.107	−0.179	<0.001	1.040
Lack of initiative ^c^								
Resilience	−0.416	−0.531 to −0.300	<0.001	−0.419	−0.535 to −0.303	−0.315	<0.001	1.040
Lack of motivation ^d^								
Resilience	−0.403	−0.527 to −0.278	<0.001	−0.396	−0.521 to −0.271	−0.279	<0.001	1.040
Quiet Quitting Scale ^e^								
Resilience	−0.309	−0.401 to −0.217	<0.001	−0.324	−0.415 to −0.232	−0.308	<0.001	1.040

^a^ Multivariable models are adjusted for gender, age, educational level, and work experience. ^b^ R^2^ for the multivariable model = 5.6%, *p*-value for ANOVA < 0.001. ^c^ R^2^ for the multivariable model = 12.6%, *p*-value for ANOVA < 0.001. ^d^ R^2^ for the multivariable model = 10.4%, *p*-value for ANOVA < 0.001. ^e^ R^2^ for the multivariable model = 12.4%, *p*-value for ANOVA < 0.001. CI: confidence interval, VIF: variance inflation factor.

**Table 5 healthcare-13-02064-t005:** Linear regression models with work engagement as the dependent variable (N = 462).

Dependent Variable Independent Variable	Univariate Models				Multivariable Model ^a^			
	Unadjusted Coefficient Beta	95% CI for Beta	*p*-Value	Adjusted Coefficient Beta	95% CI for Beta	Standardized Coefficient Beta	*p*-Value	VIF
Work engagement ^b^								
Resilience	0.501	0.310 to 0.691	<0.001	0.506	0.312 to 0.700	0.237	<0.001	1.040

^a^ Multivariable model is adjusted for gender, age, educational level, and work experience. ^b^ R^2^ for the multivariable model = 5.7%, *p*-value for ANOVA < 0.001. CI: confidence interval, VIF: variance inflation factor.

## Data Availability

The data presented in this study are available on request from the corresponding author.

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
