# Peer review of "Resilience Protects Nurses from Workplace Gaslighting and Quiet Quitting, and Improves Their Work Engagement: A Cross-Sectional Study in Greece"

_healthcare, 2025, doi:10.3390/healthcare13162064_

Round 1
Reviewer 1 Report
Comments and Suggestions for Authors
Dear Authors,
The manuscript's topic is very current and important and concerns the challenging and alarming issue of workplace gaslighting, silent resignation, and work engagement among nurses. The particular importance of the study is that it deals with identifying predictors of these phenomena in the workplace.
I would like to make suggestions for improving the manuscript:
Abstract: The abstract is structured according to the journal's instructions. However, the number of words (318) was significantly higher than the predicted 250. Consider whether you need all the facts in lines 21-22 and all the mentioned data related to the obtained results. Please correct it.
Introduction: The introduction is coherent, easy to follow, provides a comprehensive overview of the issues the manuscript addresses, and follows a funnel-shaped structure. However, for a more adequate overview of the importance of the study and the issues it deals with, in addition to the data on the global level, it would be important to present the current situation in the country where the study was conducted. For example, what are the most significant current challenges for nurses in the workplace? How many nurses are currently employed in the healthcare system, and what is their level of education? Please add this data.
Materials and Methods: In part 2.1. it remains unclear, in addition to the mentioned nurses' social networks, who the creators of those networks are, whether they are institutional networks, networks of professional associations and chambers of nursing. How did you manage to control the criteria for inclusion in the study: Participants were required to be clinical nurses in healthcare facilities, subordinates rather than supervisors. I suggest you correct/complete them. In part 2.2. and 2.4. are described adequately and in detail to replicate the study.
Results: The results are clearly presented in tables, with appropriate accompanying text.
Discussion: The discussion is extensive, and the authors draw attention to many important questions raised by their research and the results of other authors.
The conclusions are concise, well-argued, and based on research results.
The references mentioned are relevant to the topic that the paper dealt with.
I hope you find my comments helpful.

Author Response
Reviewer 1
Dear Authors,
The manuscript's topic is very current and important and concerns the challenging and alarming issue of workplace gaslighting, silent resignation, and work engagement among nurses. The particular importance of the study is that it deals with identifying predictors of these phenomena in the workplace.
I would like to make suggestions for improving the manuscript:
Comment
Abstract: The abstract is structured according to the journal's instructions. However, the number of words (318) was significantly higher than the predicted 250. Consider whether you need all the facts in lines 21-22 and all the mentioned data related to the obtained results. Please correct it.
Answer: done
Now, the abstract includes 237 words. We delete some facts in lines 21-22 and detailed results, i.e., coefficient beta and 95% confidence intervals. Please, see the updated abstract.
Comment
Introduction: The introduction is coherent, easy to follow, provides a comprehensive overview of the issues the manuscript addresses, and follows a funnel-shaped structure. However, for a more adequate overview of the importance of the study and the issues it deals with, in addition to the data on the global level, it would be important to present the current situation in the country where the study was conducted. For example, what are the most significant current challenges for nurses in the workplace? How many nurses are currently employed in the healthcare system, and what is their level of education? Please add this data.
Answer: done
We added a paragraph in the Introduction section (before Materials and Methods) regarding nursing staff in Greece.
Comment
Materials and Methods: In part 2.1. it remains unclear, in addition to the mentioned nurses' social networks, who the creators of those networks are, whether they are institutional networks, networks of professional associations and chambers of nursing. How did you manage to control the criteria for inclusion in the study: Participants were required to be clinical nurses in healthcare facilities, subordinates rather than supervisors. I suggest you correct/complete them. In part 2.2. and 2.4. are described adequately and in detail to replicate the study.
Answer: done
Dear Reviewer, thank you especially for this comment.
We add the following text.
…Nurses’ groups were institutional networks and networks of professional associations of nurses…
…Before nurses complete the study questionnaire, we asked them if they have been working as clinical nurses for at least for one year. Nurses with positive answers then were allowed to fill in the study questionnaire…
Comment
Results: The results are clearly presented in tables, with appropriate accompanying text.
Answer: Thank you for your kindness.
Comment
Discussion: The discussion is extensive, and the authors draw attention to many important questions raised by their research and the results of other authors.
The conclusions are concise, well-argued, and based on research results.
The references mentioned are relevant to the topic that the paper dealt with.
Answer: Thank you for your kindness.
I hope you find my comments helpful.
Reviewer 2 Report
Comments and Suggestions for Authors
This study addresses a timely and important issue—the role of resilience in mitigating workplace gaslighting and quiet quitting while enhancing work engagement among nurses. The topic is relevant to healthcare workforce sustainability, and the authors employ multiple validated tools with an adequate sample size. However, several methodological, conceptual, and reporting weaknesses limit the robustness and generalizability of the findings.
Major Comments:
-
Study Design Limitations:
The cross-sectional, convenience sample design precludes causal inference. While the authors note associations, some statements in the abstract and conclusion imply causality, which should be avoided or clearly qualified. -
Sampling and Representativeness:
The recruitment via an online survey may introduce self-selection bias, limiting generalizability. The paper should detail the recruitment process, inclusion/exclusion criteria, and the potential impact of this sampling method on the results. -
Operationalization of Gaslighting and Quiet Quitting:
The cultural adaptation and validation of the Gaslighting at Work Scale and Quiet Quitting Scale for the Greek context are insufficiently described. Without clear evidence of psychometric adequacy in the study population, the validity of findings is uncertain. -
Statistical Analysis Transparency:
While multivariable regression is appropriate, more detail is needed on model selection, collinearity checks, and assumptions testing. Reporting standardized coefficients and effect sizes for all associations would improve interpretability. -
Novelty and Literature Positioning:
The introduction claims novelty, but there is no comprehensive review of prior studies linking resilience to gaslighting or quiet quitting in healthcare settings. Strengthening this background will better position the study in the literature. -
Interpretation Overreach:
The discussion occasionally overstates implications for policy and practice given the cross-sectional nature of the data. Recommendations should be more cautious and grounded in the study’s limitations.
Minor Comments:
-
Clarify demographic reporting: percentages in Table 1 should sum consistently and missing data should be indicated.
-
Ensure consistent terminology: e.g., “Quiet Quitting Scale” vs. “QQS,” “gaslighting” vs. “workplace gaslighting.”
-
Figures and tables: include exact p-values where possible rather than “p<0.001” for transparency.
-
Several references lack complete bibliographic details and should conform to MDPI style.
Author Response
Reviewer 2
This study addresses a timely and important issue—the role of resilience in mitigating workplace gaslighting and quiet quitting while enhancing work engagement among nurses. The topic is relevant to healthcare workforce sustainability, and the authors employ multiple validated tools with an adequate sample size. However, several methodological, conceptual, and reporting weaknesses limit the robustness and generalizability of the findings.
Major Comments:
Comment
1. Study Design Limitations:
The cross-sectional, convenience sample design precludes causal inference. While the authors note associations, some statements in the abstract and conclusion imply causality, which should be avoided or clearly qualified.
Answer: done
Dear Reviewer, thank you especially for this comment.
To clarify this issue, we add the following text in the Abstract.
…However, the cross-sectional nature of this study cannot imply causality between the study variables, and, thus, further studies are required to clarify the association between resilience, workplace gaslighting, quiet quitting and work engagement…
Also, we add the following text in the conclusions section.
…However, as we mentioned above this study faces several limitations, and, thus, we cannot infer a causal relationship between resilience, workplace gaslighting, quiet quitting and work engagement. Further studies should reduce bias, and, therefore, improve our ability to extract valid results…
Comment
2. Sampling and Representativeness:
The recruitment via an online survey may introduce self-selection bias, limiting generalizability. The paper should detail the recruitment process, inclusion/exclusion criteria, and the potential impact of this sampling method on the results.
Answer: done
We add the following text in the section 2.1 Study design.
… Our inclusion criteria were the following: (a) participants were required to be clinical nurses in healthcare facilities, (b) nurses should be subordinates and not supervisors, (c) nurses should have at least one year of work experience, and (d) nurses should consent to participate in our study. Before nurses complete the study questionnaire, we asked them if they have been working as clinical nurses for at least for one year. Nurses with positive answers then were allowed to fill in the study questionnaire…
Also, we add the following text in the limitations section.
…Additionally, self-selection bias is probable in our study since we employed an online survey to recruit our nurses. It is probable that levels of social media use were lower among older nurses with more years of experience. For instance, mean age of our nurses was low (36.8 years). Future studies should include more representative samples to further examine the association between resilience, workplace gaslighting, quiet quitting, and work engagement…
Comment
3. Operationalization of Gaslighting and Quiet Quitting:
The cultural adaptation and validation of the Gaslighting at Work Scale and Quiet Quitting Scale for the Greek context are insufficiently described. Without clear evidence of psychometric adequacy in the study population, the validity of findings is uncertain.
Answer: done
We add the following text in the section 2.2 Measurements.
…The Greek version of the QQS has already proven reliable and valid in a sample of nurses [39]. In short, confirmatory factor analysis confirmed the three-factor nine-item structure of the QQS since all goodness-of-fit statistics have acceptable values. Also, all Cronbach’s alpha and McDonald’s Omega values were higher than 0.70. Moreover, Cohen’s kappa for the nine items ranged from 0.840 to 0.947, while intraclass correlation coefficients for the QQS and the three subfactors ranged from 0.972 to 0.988….
….The Greek version of the GWS is a reliable and valid tool since a study with 400 participants confirmed the two-factor 11-item structure of the scale through factor analysis. Additionally, the GWS showed very good concurrent validity through moderate to high correlations with four other valid scales. Cronbach’s alpha and McDonald’s Omega for the GWS were 0.939 and 0.949, respectively. Additionally, test-retest study showed very high intraclass correlation coefficients and Cohen’s kappa values...
Comment
4. Statistical Analysis Transparency:
While multivariable regression is appropriate, more detail is needed on model selection, collinearity checks, and assumptions testing. Reporting standardized coefficients and effect sizes for all associations would improve interpretability.
Answer: done
We check for multicollinearity and assumptions to perform the multivariable linear regression analysis. Also, we report standardized coefficients betas. Since there are now 16 figures, we present them in supplementary material. Please, see the supplementary material. Also, we add standardized coefficients betas and variance inflation factors in the Tables. Please, see the Tables.
Also, we add the following text in the section 2.4 Statistical analysis.
…Moreover, we used variance inflation factors (VIFs) to assess multicollinearity in the multivariable models. A VIF greater than 5 indicates multicollinearity between independent variables. Additionally, we examined histograms of the residuals to check for multivariable normality. We examined scatterplots of residuals versus predicted values to check for homoscedasticity and linearity…
Additionally, we add the following text in the Results section.
Figure S1 indicates multivariable normality for the multivariable model with loss of self-trust as the dependent variable since the residuals followed a normal distribution. Figure S2 indicates homoscedasticity and linearity of the multivariable model, with loss of self-trust as the dependent variable. VIFs for the final multivariable model ranged from 1.040 to 1.060, indicating absence of multicollinearity between independent variables.
Figures S3 and S4 refer to multivariable model with abuse of power as the dependent variable. Figures indicate normality, homoscedasticity and linearity of the multivariable model. Moreover, VIFs for the final multivariable model ranged from 1.040 to 1.060, indicating absence of multicollinearity between independent variables.
Figure S5 denotes multivariable normality for the multivariable model with workplace gaslighting score as the dependent variable since the residuals followed a normal distribution. Also, we found that the multivariable model shows homoscedasticity and linearity (Figure S6). There was no multicollinearity between independent variables since VIFs ranged from 1.040 to 1.060.
Figures S7 and S8 refer to multivariable model with detachment as the dependent variable. There was evidence for normality, homoscedasticity and linearity of the multivariable model.
Figure S9 indicates multivariable normality for the multivariable model with lack of initiative as the dependent variable since the residuals followed a normal distribution. Also, we found that the multivariable model shows homoscedasticity and linearity (Figure S10).
There was multivariable normality for the multivariable model with lack of motivation as the dependent variable (Figure S11). Also, scatterplot of residuals versus predicted values with lack of motivation as the dependent variable indicated homoscedasticity and linearity of the multivariable model (Figure S12).
Figure S13 indicates multivariable normality for the multivariable model with QQS score as the dependent variable since the residuals followed a normal distribution. Figure S14 indicates homoscedasticity and linearity of the multivariable model, with QQS score as the dependent variable.
VIFs for all final multivariable models ranged from 1.040 to 1.060, indicating absence of multicollinearity between independent variables in all cases.
Figure S15 indicates multivariable normality for the multivariable model with work engagement score as the dependent variable since the residuals followed a normal distribution. Also, we found that the multivariable model shows homoscedasticity and linearity (Figure S16). There was no multicollinearity between independent variables since VIFs ranged from 1.040 to 1.060.
5. Novelty and Literature Positioning:
The introduction claims novelty, but there is no comprehensive review of prior studies linking resilience to gaslighting or quiet quitting in healthcare settings. Strengthening this background will better position the study in the literature.
Answer: This is the first study in the literature that investigated the association between resilience and workplace gaslighting and quiet quitting.
6. Interpretation Overreach:
The discussion occasionally overstates implications for policy and practice given the cross-sectional nature of the data. Recommendations should be more cautious and grounded in the study’s limitations.
Answer: Done
We added a paragraph, before limitations, regarding recommendations grounded in the study’s limitations
Minor Comments:
Comment
1. Clarify demographic reporting: percentages in Table 1 should sum consistently and missing data should be indicated.
Answer: done
We check Table 1. Sum of percentages for each variable is 100.0%. There were no missing data. We add this information in statistical analysis section. We add the following text.
… There was no missing data…
2. Ensure consistent terminology: e.g., “Quiet Quitting Scale” vs. “QQS,” “gaslighting” vs. “workplace gaslighting.”
Answer: Done
We retained the term workplace gaslighting only where it appeared in the measurement tool.
Comment
3. Figures and tables: include exact p-values where possible rather than “p<0.001” for transparency.
Answer: done
Dear Reviewer, please take into your consideration that p-values are extremely low. For instance, 0.000000004647 and 0.00000000004067. This is the reason that we present these p-values as <0.001. Please, let us know if you want to replace p<0.001 with the exact p-values.
4. Several references lack complete bibliographic details and should conform to MDPI style.
Round 2
Reviewer 2 Report
Comments and Suggestions for Authors
The revised manuscript has taken into account my feedback.